# Evolution of the potassium channel gene *Kcnj13* underlies colour pattern diversification in *Danio* fish

Marco Podobnik [1], Hans Georg Frohnhöfer[1], Christopher M. Dooley[1,2], Anastasia Eskova[1,3], Christiane Nüsslein-Volhard[1] & Uwe Irion [1✉]

The genetic basis of morphological variation provides a major topic in evolutionary developmental biology. Fish of the genus *Danio* display colour patterns ranging from horizontal stripes, to vertical bars or spots. Stripe formation in zebrafish, *Danio rerio*, is a self-organizing process based on cell–contact mediated interactions between three types of chromatophores with a leading role of iridophores. Here we investigate genes known to regulate chromatophore interactions in zebrafish that might have evolved to produce a pattern of vertical bars in its sibling species, *Danio aesculapii*. Mutant *D. aesculapii* indicate a lower complexity in chromatophore interactions and a minor role of iridophores in patterning. Reciprocal hemizygosity tests identify the potassium channel gene *obelix/Kcnj13* as evolved between the two species. Complementation tests suggest evolutionary change through divergence in *Kcnj13* function in two additional *Danio* species. Thus, our results point towards repeated and independent evolution of this gene during colour pattern diversification.

[1] Max Planck Institute for Developmental Biology, Max-Planck-Ring 5, 72076 Tübingen, Germany. [2]Present address: Max Planck Institute for Heart and Lung Research, Ludwigstrasse 43, 61231 Bad Nauheim, Germany. [3]Present address: IBM Research and Development, Schönaicher Straße 220, 71032 Böblingen, Germany. ✉email: uwe.irion@tuebingen.mpg.de

Colour patterns are common features of animals and have important functions in camouflage, as signals for kin recognition, or in mate choice. As targets for natural and sexual selection, they are of high evolutionary significance[1–4]. Colour patterns are highly variable and evolve rapidly leading to large diversities even within a single genus and to remarkable similarities in distant genera. The patterns frequently involve spots or stripes of different orientations. The identification of genes involved in colour patterning has become a major goal in evolutionary developmental biology[5–9]. Colour pattern development and evolution is studied in many systems, from insects to vertebrates, that use fundamentally different mechanisms to form the patterns. In insects most colour patterns are generated in the two-dimensional sheets of epidermal cells that produce pigments or light-scattering structures, which are secreted into the cuticle. The patterns often are oriented along morphological landmarks, such as segment boundaries or wing veins. Patterning in butterfly wings is essentially controlled by well-known signalling systems such as *dpp* or *hedgehog* and guided by spatially expressed transcription factors serving as anatomical prepatterns[10]. Particularly well studied are the wing patterns of *Heliconius* butterflies where adaptive radiations in Central and South America led to many species with a large variety of patterns that are used in Müllerian mimicry and predator avoidance. It has been found that only few genes with large effects cause wing pattern adaptations in these species; cis-regulatory changes in the gene, *optix*, were identified as the basis for the convergent evolution of the patterns in a number of different species[11,12].

In contrast to insects, pigment patterns in vertebrates are not of epithelial origin, but are produced by specialised pigment cells (chromatophores) derived from the neural crest, a transient embryonic structure that develops at the boundary between neural tissue and epidermis. The neural crest cells delaminate from the invaginating neural tube, become migratory and distribute in the organism contributing to many different tissues and organs. The pigment cell precursors migrate into the skin where they distribute and produce pigments or structural colours. Whereas mammals and birds only possess one type of pigment cell, the melanocyte producing brown or black melanin pigments, several more pigment cell types are present in cold-blooded vertebrates such as fish, amphibia and reptiles; most widely distributed are orange to yellow xanthophores, red erythrophores and light reflecting white or silvery iridophores[13]. Differential distribution and superposition of pigment cells allows for the generation of a rich diversity of colour patterns in these basal vertebrates. Pattern formation by neural crest-derived pigment cells involves direct contact-based interactions among cells of the same type or between different types of pigment cells. These interactions control cell proliferation, shape changes and migration resulting in superimposed layers of differently coloured pigment cells under the skin generating a large variety of patterns, particularly rich in fishes.

The adult patterns of fish as targets for sexual selection and kin recognition are particularly well suited to study the evolution of colour patterns in vertebrates: In many genera a rich diversity of patterns in closely related species exist, and the development of the adult patterns in the juvenile fish can be followed directly as it takes place outside the maternal organism. A teleost-specific whole genome duplication followed by sub-functionalization of the paralogues resulted in many genes in fish that are specific for adult colour patterning without having other vital functions, thus reducing constraints for the evolution of these genes[14,15]. Cichlids from the great African lakes are examples of recent adaptive radiations that led to the emergence of hundreds of new species and sub-species with many divergent patterns, frequently made up of bars or horizontal stripes of different colours. Genetic mapping using hybrids between striped and non-striped cichlid species was recently used to show that the secreted signalling molecule Agouti-related peptide 2 (Agrp2) is a main driver in the suppression of horizontal stripes[16]. Further analysis revealed higher levels of expression of *Agrp2* in other non-striped species compared to striped species from two different lake systems, confirming a further example of convergent evolution of the same gene.

The zebrafish, *Danio rerio*, has emerged as an excellent system to study colour pattern development in a vertebrate[7,8,13,17–19]. In this model organism a fair number of genes have been identified in mutant screens that are required for the formation of the pattern[7,8], which is composed of a series of horizontal light and dark stripes on the flank of the fish as well as in the anal and tail fins (Fig. 1a). The adult pattern is created by three different types of chromatophores in the skin, in the dark stripes black melanophores are overlaid by blue iridophores and lightly coloured stellate xanthophores. The light stripes are composed of dense, silvery iridophores underneath compact orange xanthophores[20–23]. The chromatophores producing this pattern mainly originate from multipotent neural crest-derived stem cells located at the dorsal root ganglia of the peripheral nervous system[24–28]. Several signalling pathways control proliferation and tiling of the different chromatophore types; Kit-signalling is required for most melanophores, Csf1-signalling for the development of xanthophores and Edn3-signalling for dense iridophores[29–31]. During metamorphosis, the period when adult form and colour pattern are established, stripe formation is initiated by iridophores emerging along the horizontal myoseptum. Iridophores proliferate and spread in the skin to form a series of light stripes alternating with dark stripes of melanophores that emerge in between. Cell shape changes and assembly into the striped pattern are controlled by interactions among the three cell types[23,32–34]. Several genes are autonomously required in the chromatophores for these heterotypic interactions[32,33,35–40]. These genes typically encode integral membrane proteins such as adhesion molecules[36], channels[38], or components of cellular junctions, some of which mediate direct cell contacts[40–42]. Stripe formation is also influenced by the local tissue environment[43–45] and by global hormonal signals, such as galanin-regulated thyroid hormone[46–48] and insulin[49]. The correct orientation of the stripes in zebrafish depends on the horizontal myoseptum. In *Meox1* (*choker*) mutants, which lack this structure, the horizontal orientation of the stripes is lost, but they form of normal width and composition (Fig. 1b), indicating that stripe formation is a process of self-organisation of the pigment cells[32].

To study the evolution of colour patterns we can now, based on the knowledge we have from the model organism *D. rerio*, examine other closely related *Danio* species. These show an amazing variety of colour patterns, which range from horizontal stripes in *D. rerio* (Fig. 1a), to vertical bars in *D. aesculapii*[50], *D. choprae* or *D. erythromicron* (Fig. 1c, g, m), spotted patterns in *D. tinwini*[51] or *D. margaritatus* (Fig. 1d, h) or an almost complete lack of any pattern in *D. albolineatus*. The *Danio* species diversified for at least 13 million years in Southeast Asia and their spatial distributions only partially overlap today[52,53]. Hybrids between *D. rerio* and other *Danio* species can be produced in the laboratory by natural matings or by in vitro fertilisation[54]. They invariably display colour patterns similar to the stripes in *D. rerio*, thus, horizontal stripes appear to be dominant over divergent patterns (Fig. 1e, f, i, j)[54]; whether this is due to a gain-of-function in striped species or losses in the other species is an open question[7,8,18]. The hybrids are virtually sterile impeding further genetic experiments, like QTL mapping, but they allow interspecific complementation tests[54].

Three *Danio* species, *D. aesculapii*, *D. choprae* and *D. erythromicron*, display vertical bars. Surprisingly, these species are

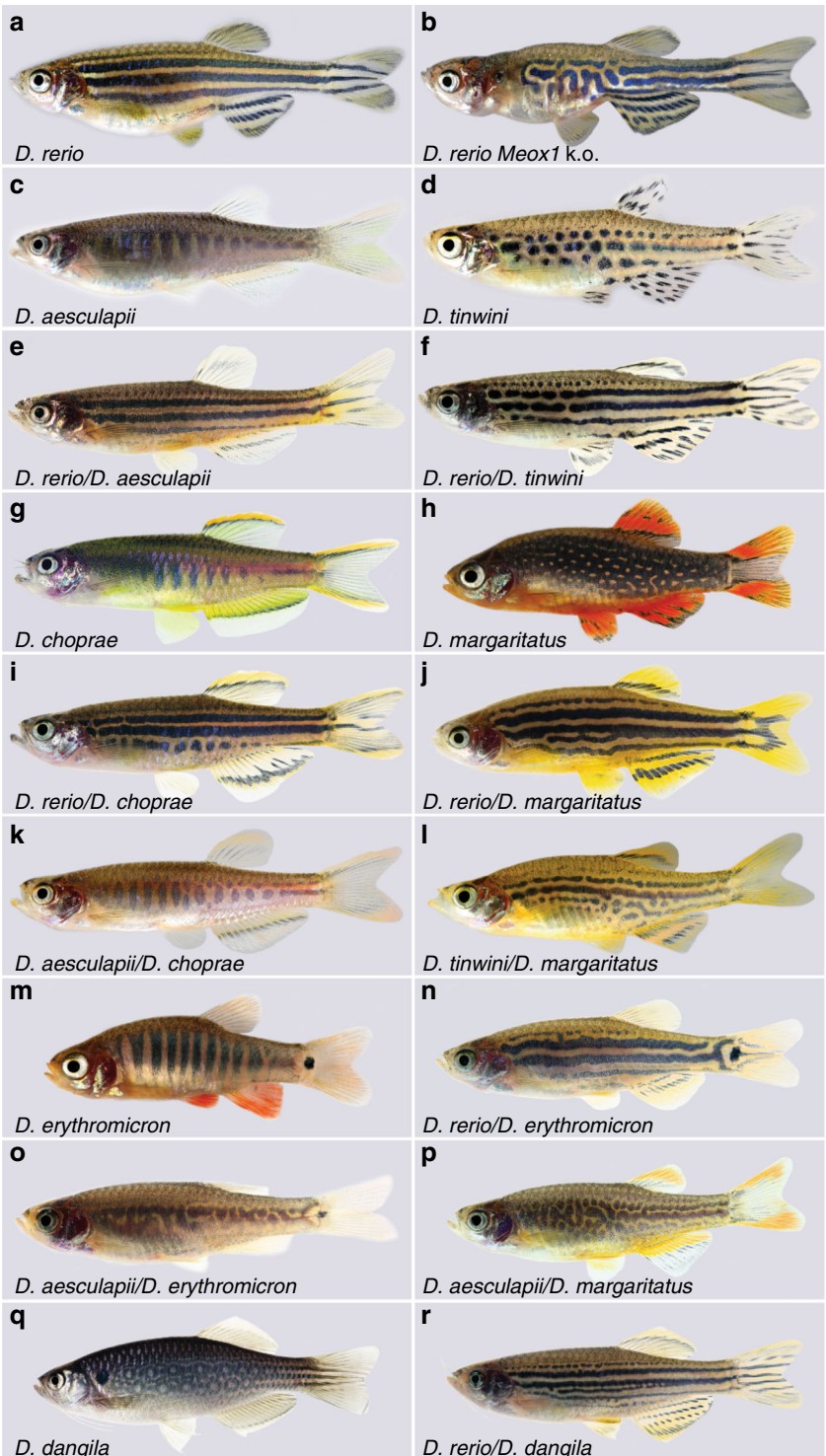

**Fig. 1 Colour patterns in *Danio* fish and interspecific hybrids. a** Colour pattern of zebrafish, *D. rerio*. **b** *D. rerio Meox1* (*choker*) mutants, which lack a horizontal myoseptum. **c** *D. aesculapii*. **d** *D. tinwini*. **e** Hybrid between *D. rerio* and *D. aesculapii* and **f** hybrid between *D. rerio* and *D. tinwini*. **g** *D. choprae*. **h** *D. margaritatus*. **i** Hybrid between *D. rerio* and *D. choprae*, **j** and hybrid between *D. rerio* and *D. margaritatus*. **k** Hybrid between *D. aesculapii* and *D. choprae*. **l** Hybrid between *D. tinwini* and *D. margaritatus*. **m** *D. erythromicron*. **n** Hybrid between *D. rerio* and *D. erythromicron*. **o** Hybrid between *D. aesculapii* and *D. erythromicron*. **p** Hybrid between *D. aesculapii* and *D. margaritatus*. **q** *D. dangila*. **r** Hybrid between *D. rerio* and *D. dangila*. All pictures are representative for the corresponding species or hybrids; for the variability of hybrid patterns see Supplementary Fig 1. Please note that not all panels are shown to the same scale; the sizes of the fish are ~18 mm (*D. margaritatus* and *D. erythromicron*), 24 mm (*D. tinwini*), 30 mm (*D. choprae*), 35 mm (*D. rerio*, *D. aesculapii*) and 75 mm (*D. dangila*).

not monophyletically related within the genus *Danio*. In this paper we describe hybrids between these barred species, showing that all three of them lack the cues for a horizontal orientation of the pattern. However, we find that vertical bars develop in a different manner in *D. erythromicron* compared to *D. aesculapii* and *D. choprae*, showing that bar formation likely evolved convergently by two different modes.

Using the CRISPR/Cas9 system, we generated loss-of-function mutations in known regulators of chromatophore interactions from *D. rerio* in its closest sibling species, *D. aesculapii*. The phenotypes confirmed that these genes regulate patterning also in this species and demonstrate a lower complexity in the interactions among chromatophores. Further they suggest a minor role of iridophores in the patterning of this barred species compared to *D. rerio*[32,33]. We then performed reciprocal hemizygosity tests[55] with null alleles of four known regulators of chromatophore interactions, the two connexin genes *Cx39.4* (*luchs*)[40] and *Cx41.8/Gja5b* (*leopard*)[37,41], the potassium channel gene *Kcnj13* (*obelix/jaguar*)[37,38] and the cell adhesion gene *Igsf11* (*seurat*)[36]. In the case of *Kcnj13*, we found that the reciprocal hybrids display qualitatively different phenotypes indicating that the function has diverged between *D. rerio* and *D. aesculapii*, whereas in the other three cases the function of the genes appears to be conserved. One-way complementation tests with eight more *Danio* species suggest that the *Kcnj13* gene has also evolved between *D. rerio* and two more species, *D. tinwini* and *D. choprae*. The separated phylogenetic positions of these species suggest that the evolution of *Kcnj13* contributing to the pattern diversity in *Danio* fish has occurred several times independently.

## Results

**Horizontal pattern orientation is lost in barred species.** To reconstruct the history of colour pattern evolution we first investigated how pattern orientation is inherited in hybrids. The horizontal orientation of the stripes in *D. rerio* (Fig. 1a) depends on the horizontal myoseptum along which iridophores emerge to form the first light stripe. In *Meox1* mutants (*choker*), which lack the horizontal myoseptum, meandering stripes form without clear orientation (Fig. 1b)[32]. The closest sibling species to *D. rerio*, *D. aesculapii*, shows a very different pattern of vertically oriented dark bars (Fig. 1c)[50]. Similar barred patterns are exhibited by the more distantly related *D. choprae* and *D. erythromicron* (Fig. 1g, m). These patterns clearly do not use the horizontal myoseptum, which is present in all species, for orientation. In all three cases, hybrids with *D. rerio* show a pattern that resembles the horizontal *D. rerio* stripes (Fig. 1e, i, n)[8]. Strikingly, hybrids between *D. aesculapii* and *D. choprae* display a barred pattern, similar to the species pattern (Fig. 1k). This indicates that in both species the cues for horizontal orientation are lacking and that the barred pattern develops in a similar manner. In contrast, hybrids between *D. aesculapii* and *D. erythromicron* develop highly variable patterns without any clear orientation (Fig. 1o and Supplementary Fig. 1). Therefore, the vertical bars must develop in a different manner in *D. erythromicron* compared to *D. aesculapii* and *D. choprae*.

Two *Danio* species display spotted patterns: *D. tinwini* has dark spots on a light background (Fig. 1d)[51], whereas *D. margaritatus* shows light spots on a dark background (Fig. 1h). In both cases, hybrids with *D. rerio* show a stripe pattern similar to *D. rerio* (Fig. 1f, j)[8]. Hybrids between the two spotted species also develop a pattern of horizontal stripes, albeit with some interruptions and irregularities (Fig. 1l). These results indicate that the horizontal myoseptum functions to orient the pattern in the hybrids between *D. tinwini* and *D. margaritatus*, and therefore in at least one of the two parental species. It seems

likely that this is the case in *D. tinwini*, as the spots show some horizontal orientation reminiscent of interrupted stripes. Hybrids between *D. aesculapii* and *D. margaritatus* develop meandering patterns that do not resemble either of the parental species and lack a clear horizontal or vertical orientation (Fig. 1p). *D. dangila*, the most distantly related species to *D. rerio* that we examined in this study, show a pattern of rows of dark rings (Fig. 1q). Hybrids between *D. rerio* and *D. dangila* develop horizontal stripes, which often partially split (Fig. 1r)[54]. Based on the most recent phylogeny[52], we hypothesise an evolutionary history, in which the horizontal orientation of the pattern in the *D. rerio* group was gained from an ancestral ambiguous pattern and lost again in *D. aesculapii*. Two other species, *D. erythromicron* and *D. choprae*, independently might have acquired a vertical orientation from this ancestral pattern. The patterns of the hybrids between *D. aesculapii* and *D. erythromicron* or *D. margaritatus*, which are more variable than the species patterns (Supplementary Fig. 1) and without clear orientation, might resemble such an ancestral pattern. A variable ancestral pattern without well-defined orientation might not have functioned as recognition signal but rather provided camouflage.

**A minor role of iridophores in cellular interactions forming bars.** To investigate the developmental and genetic basis for the differences in pattern orientation, we focussed on the sibling species *D. rerio* and *D. aesculapii*, which display horizontal stripes and vertical bars, respectively (Fig. 1a, c). In *D. rerio*, during early metamorphosis, iridophores emerge along the horizontal myoseptum to form the first light stripe (Fig. 2a)[20,32,33]. In contrast, in *D. aesculapii* iridophores appear only during later stages, are more scattered over the flank and fewer in number (Fig. 2b, d). This indicates that it is not the physical presence of the horizontal myoseptum, which exists in both species, but rather specific guidance signals, which are not present in *D. aesculapii*, that direct iridophores into the skin in *D. rerio*. Later, when iridophores, covered by compact xanthophores, have formed the first contiguous light stripe with adjacent melanophore stripes in *D. rerio* (Fig. 2c, e), in *D. aesculapii* melanophores and xanthophores intermix broadly (Fig. 2f); they sort out loosely into vertical bars of low contrast without coherent sheets of dense iridophores between the melanophore bars during later stages (Fig. 2h), when the *D. rerio* pattern is already fully formed (Fig. 2g). Our observations suggest that the different patterns in these sibling species are produced by the presence or absence of guidance signals for iridophores along the horizontal myoseptum as well as by cellular interactions that prevent mixing of melanophores and xanthophores in *D. rerio* but not in *D. aesculapii*.

To address the role of the different cell types, we used the CRISPR/Cas9 system to generate mutants lacking individual chromatophore types in *D. aesculapii*. Whereas in *D. rerio* vestiges of the striped pattern form in the absence of one chromatophore type (Fig. 3a, c, e)[32], loss of either melanophores (Fig. 3b) or xanthophores (Fig. 3d) completely abolishes the patterning in *D. aesculapii*. This indicates that the repulsive interactions between melanophores or xanthophores and iridophores, which account for the residual patterns in *D. rerio*[32,33], are absent in *D. aesculapii*. In contrast, eliminating iridophores in *D. aesculapii* still permits some melanophore bar formation (Fig. 3f). This indicates that iridophores, which play a dominant role for stripe formation in *D. rerio*, are dispensable for the formation of vertical bars in *D. aesculapii*.

**Weak heterotypic chromatophore interactions in bars.** Next, we analysed genes with known functions in heterotypic interactions between chromatophores in *D. rerio*. Null alleles in the connexin

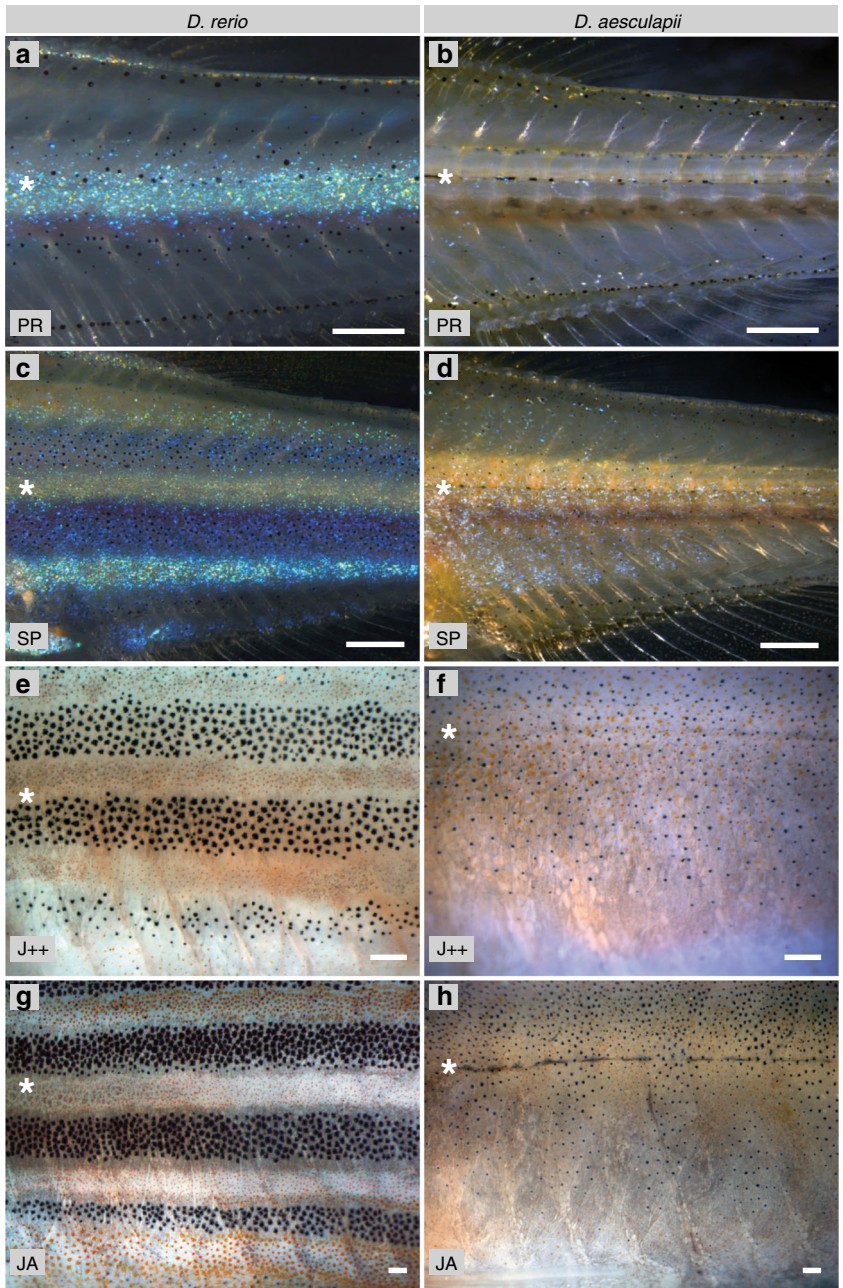

**Fig. 2 Development of colour patterns in *D. rerio* and *D. aesculapii*. a** *D. rerio* fish at stage PR, iridophores (arrowhead) emerge along the horizontal myoseptum (asterisk) to form the first light stripe. **b** *D aesculapii* fish at stage PR. **c** *D. rerio* at stage SP, the first light stripe is flanked dorsally and ventrally by emerging dark stripes. **d** *D. aesculapii* at stage SP, iridophores emerge in a scattered fashion. **e** *D. rerio* at stage J++, light stripes are covered by compact xanthophores. **f** *D. aesculapii* at stage J++, melanophores and xanthophores broadly intermix. **g** *D. rerio* at stage JA, the stripes are fully formed. **h** *D. aesculapii* at stage JA, melanophores and xanthophores sort out loosely into vertical bars of low contrast; no dense iridophores are visible between the dark bars. **a–d** Incident light illumination to highlight iridophores, **e–h** bright field illumination to visualise xanthophores and melanophores. All pictures are representative for the corresponding species and stages ($n \geq 3$). Staging of animals according to Parichy et al.[83]. PB (pectoral fin bud, 7.2 mm SL). SP (squamation posterior, 9.5 mm SL). J++ (juvenile posterior, 16 mm SL). JA (juvenile-adult, >16 mm SL). Scale bars correspond to 250 μm.

genes *Cx39.4* (*luchs*)[40] and *Cx41.8/Gja5b* (*leopard*)[37,41] lead to melanophore spots (Fig. 4a, b). Both connexins are thought to form heteromeric gap junctions involved in the interaction between xanthophores and melanophores[40,42]. Missense mutations in *Igsf11* (*seurat*)[36], which codes for a cell adhesion molecule, also cause a spotted pattern. We generated a frame-shift mutation in exon 3 of *Igsf11* in *D. rerio*. This mutation leads to a truncation of the protein at the end of the first Ig-domain and is, presumably, a functional null allele. Fish heterozygous for this mutation show no mutant phenotype, whereas homozygous fish

display slightly stronger pattern aberrations than those seen in the previously identified alleles (Fig. 4c)[36].

Mutations in *Kir7.1/Kcnj13* (*obelix/jaguar*), which codes for an inwardly rectifying potassium channel, result in fewer and wider stripes with some mixing of melanophores and xanthophores[35,37,38]. So far, five *Kcnj13* alleles, all of which are dominant, have been identified in *D. rerio* in several independent genetic screens (Supplementary Figs. 4 and 5)[37,38,40,56,57]. We used the CRISPR/Cas9 system to generate novel mutations in *Kcnj13* in *D. rerio*. A six-base pair in-frame deletion in exon 1,

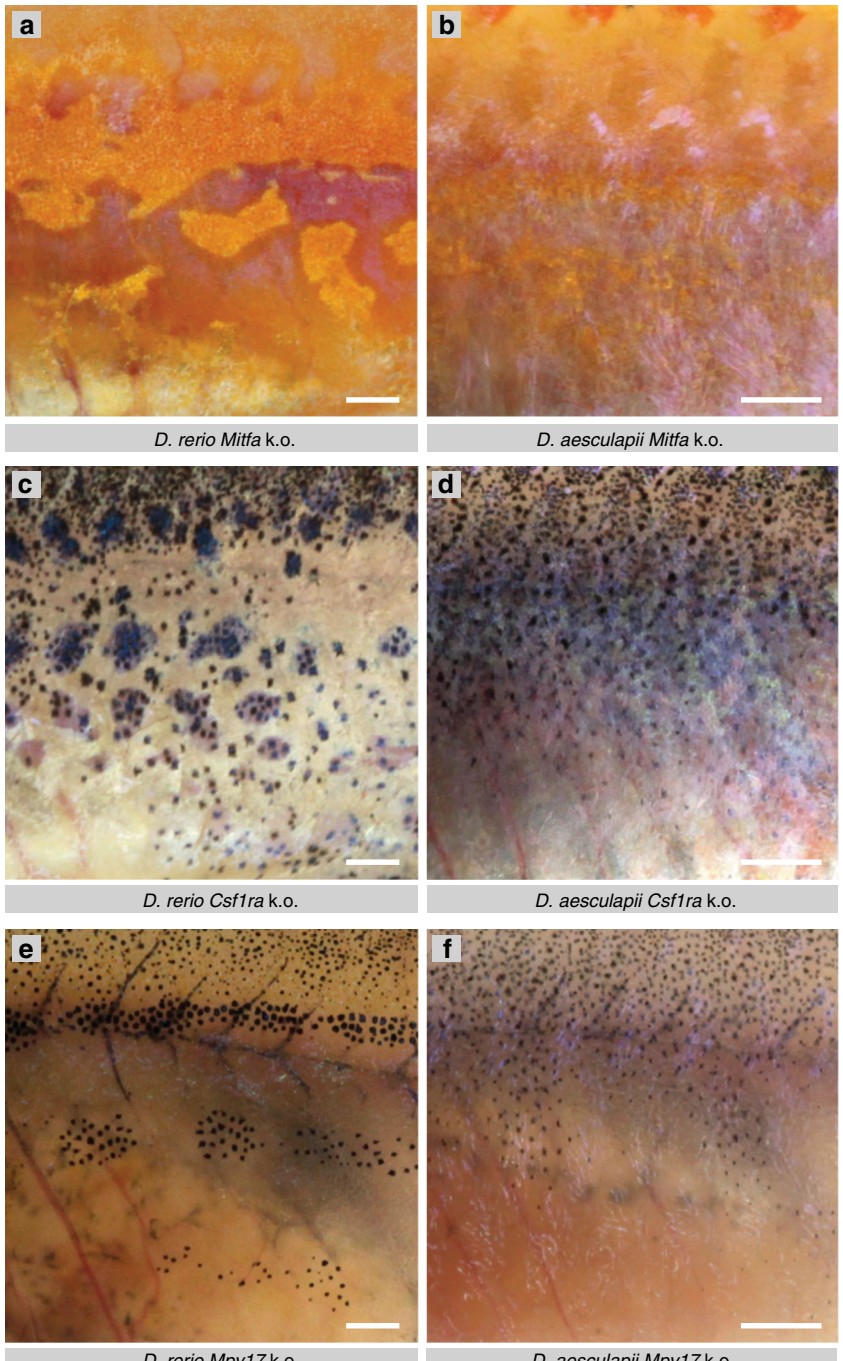

**Fig. 3 Mutant phenotypes in *D. rerio* and *D. aesculapii* of genes required for individual pigment cell types.** In *D. rerio* loss of one type of pigment cell type, **a** melanophores in *Mitfa* (*nacre*) mutants, **c** xanthophores in *Csf1ra* (*pfeffer*) mutants, or, **e** iridophores in *Mpv17* (*transparent*) mutants, still permits rudimentary aggregation of dense iridophores (**a**) or melanophores (**b**, **c**). In *D. aesculapii*, loss of melanophores, **b**, in *Mitfa* mutants ($n > 100$) or loss of xanthophores, **d**, in *Csf1ra* mutants ($n > 50$), abrogate any residual pattern formation. However, vertical melanophore bars still form in *Mpv17* mutants ($n = 8$), **f** despite the absence of iridophores. All images show representative examples of the corresponding genotypes. Scale bars correspond to 1 mm.

which leads to a loss of two amino acids in the protein (Supplementary Fig. 4), also gives rise to a dominant phenotype, similar to the already known alleles. However, using a second CRISPR target site we also recovered a frame-shift mutation. This 14-base pair insertion near the end of the first coding exon leads to an early truncation of the protein before the second transmembrane domain. This allele is recessive: heterozygous carriers have a complete wild-type pattern, homozygous mutants are indistinguishable from homozygous mutants for any of the

dominant alleles (Fig. 4d). We consider this new recessive allele to be a functional null allele.

To investigate the functions of all four genes in *D. aesculapii*, we generated presumptive null alleles in the orthologs. In all of the mutants, bar formation is abolished and we find an even distribution of melanophores (Fig. 4e–h) indicating that the interactions mediated by each of these genes are essential to generate the melanophore bars in *D. aesculapii*. The complete loss of a pattern in single mutants in *D. aesculapii* is different from *D.*

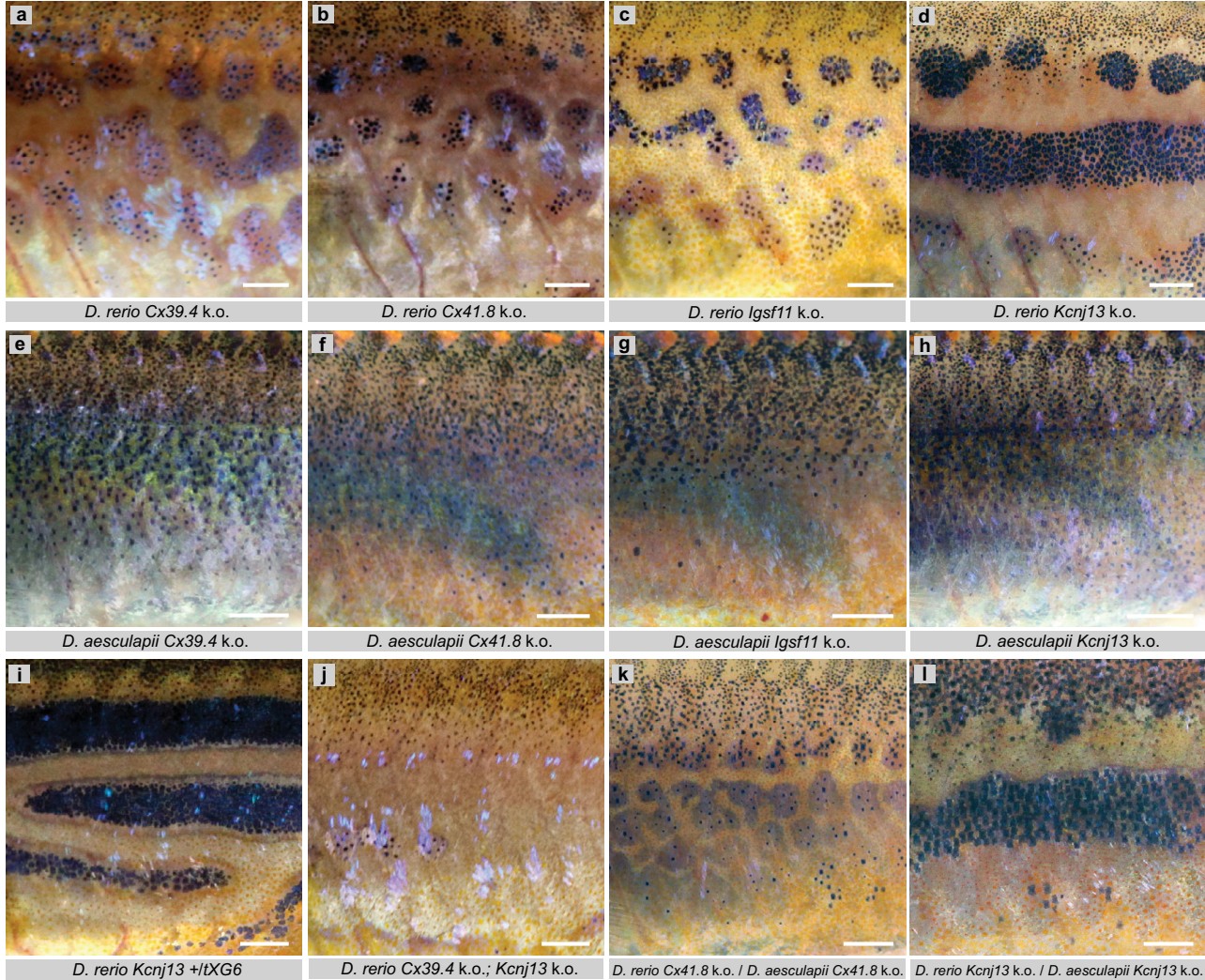

**Fig. 4 Mutant phenotypes in *D. rerio*, *D. aesculapii* and their hybrids of genes required for heterotopic interactions.** In *D. rerio* mutations in, **a**, *Cx39.4* (*luchs*), **b**, *Cx41.8* (*leopard*), and **c**, *Igsf11* (*seurat*) lead to spotted patterns, whereas, **d**, mutations in *Kcnj13* (*obelix*) result in fewer and wider stripes. In *D. aesculapii*, **e–h**, mutations in the orthologous genes lead to the complete loss of any pattern. In *D. rerio* dominant alleles of *Kcnj13*, **i**, cause broader stripes and irregularities when heterozygous. Double mutants, **j**, *Cx39.4* k.o.; *Kcnj13* k.o., loose almost all melanophores and pattern. Interspecific hybrids between *D. rerio* and *D. aesculapii*, which are both mutant, **k**, for *Cx41.8*, or, **l**, for *Kcnj13*, show patterns of spots or wider stripes similar to the corresponding *D. rerio* mutants (**b**, **d**; *n* = 15). All images show representative examples of the corresponding genotypes. Scale bars correspond to 1 mm.

*rerio* where this occurs only in double mutants (Fig. 4j)[40]. In concert with predictions of agent-based models of patterning[58], this indicates that the robust formation of horizontal stripes in *D. rerio* is due to a gain in complexity based on partially redundant chromatophore interactions. These are dominated by iridophores and oriented by an as yet unidentified signal along the horizontal myoseptum. *D. aesculapii* might have secondarily lost the dominance of iridophores leading to a pattern based primarily on interactions between xanthophores and melanophores and thus of lower complexity.

We next investigated whether these genes function in the *D. rerio*/ *D. aesculapii* hybrids in the same way they function in *D. rerio*. The ability to produce frame-shift mutations in both species allowed us to generate mutant hybrids that carry null alleles from both parental species. Wild-type hybrids form stripes similar to *D. rerio* (Fig. 1e) [8,54], hence we expect similar phenotypes comparing *D. rerio* mutants and mutant hybrids. The mutant hybrids show indeed patterning phenotypes very similar to the respective *D. rerio* mutants (Fig. 4k, l), confirming that stripe formation in the hybrids is very similar to the

process in *D. rerio* and showing that these genes have the same functions in *D. rerio* and in the hybrids.

**Kcnj13 has evolved between *D. rerio* and *D. aesculapii*.** Next, we generated reciprocal hemizygotes, i.e., interspecific hybrids carrying a null allele from each parental species in an otherwise identical genetic background[55]. We expect similar patterns in these hybrids if the gene function has not evolved between species. A qualitatively altered hybrid pattern would reveal that one of the parental alleles cannot complement the induced loss-of-function of the other, therefore indicating functional changes in the gene during evolution. We found that hemizygous hybrids with the null allele of *Kcnj13* from *D. rerio* display a novel pattern of spots or interrupted stripes whereas a striped pattern forms with the null allele from *D. aesculapii* (Fig. 5a, b and Supplementary Fig. 2a, b). This indicates a functional diversification between the wild-type alleles from *D. rerio* and *D. aesculapii*. The phenotype of the hemizygous hybrid with a functional *D. aesculapii* allele (Fig. 5a) is qualitatively different from that of the

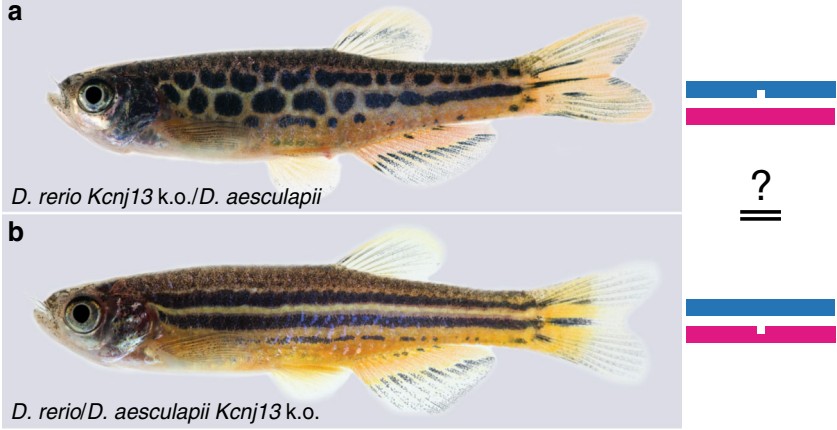

**Fig. 5 A reciprocal hemizygosity test to identify *Kcnj13* evolution.** Two hybrids between *D. rerio* and *D. aesculapii*, which are hemizygous for a *Kcnj13* loss-of-function mutation. **a** stripes are interrupted in hybrids carrying the mutant allele from *D. rerio* (*n* > 60, nick in the blue line representing the zebrafish genome). **b** hybrids carrying the mutant allele from *D. aesculapii* (*n* = 6, nick in the magenta line, representing the *D. aesculapii* genome) are indistinguishable from wild-type hybrids (Fig. 1e).

homozygous mutant hybrid (Fig. 4l), indicating that the *D. aesculapii Kcnj13* gene is functional in the hybrid. In contrast, in the case of hemizygous hybrids with *Cx39.4*, *Cx41.8* and *Igsf11* striped patterns indistinguishable from wild-type hybrids are formed regardless whether the null allele stems from *D. rerio* (Supplementary Fig. 2c, e, g) or *D. aesculapii* (Supplementary Fig. 2d, f, h). These reciprocal hemizygosity tests indicate that *Cx39.4*, *Cx41.8* and *Igsf11* provide similar functions in both species, whereas the function of *Kcnj13* has evolved between the two species.

**_Kcnj13_ may have evolved repeatedly in the _Danio_ genus.** To investigate if *Kcnj13* underlies the pattern variation more broadly across the *Danio* genus, we crossed the *D. rerio Kcnj13* null allele to eight other *Danio* species (Fig. 6 and Supplementary Fig. 3). As mentioned above, wild-type hybrids between *D. rerio* and other *Danio* species display horizontal stripes, resembling the *D. rerio* pattern, with slight defects in *D. albolineatus* (Fig. 6). Strikingly, not only *D. rerio Kcnj13*/*D. aesculapii* hybrids (Fig. 6, highlighted in magenta, Supplementary Fig. 2a) but also *D. rerio Kcnj13*/*D. tinwini* hybrids (Fig. 6, highlighted in yellow, Supplementary Fig. 3c) and *D. rerio Kcnj13* k.o./*D. choprae* hybrids (Fig. 6, highlighted in cyan, Supplementary Fig. 3e) developed patterns of spots or interrupted stripes suggesting that the *Kcnj13* function has evolved compared to *D. rerio*. As we do not yet have the means to generate reciprocal hybrids with these additional two species, we cannot completely rule out that effects of the novel genetic background in these hybrids also contribute to the observed phenotypes. The spotted pattern in the hybrids carrying the *D. rerio Kcnj13* null allele, which is qualitatively different from all wild-type hybrids and also from the *D. rerio Kcnj13* mutant pattern, is similar to the parental pattern of *D. tinwini*, where dense iridophores interrupt the dark melanophore stripes (Figs. 1d and 6). No qualitative differences were detected between wild-type hybrids and hybrids hemizygous for *D. rerio Kcnj13* in the case of *D. kyathit*, *D. nigrofasciatus*, *D. albolineatus*, *D. erythromicron*, *D. margaritatus* and *D. dangila* (Fig. 6). This indicates that the alleles from these species complement the loss of the *D. rerio Kcnj13* allele and supports the notion that the barred pattern in *D. erythromicron* develops in a different manner from the other two barred species. Taken together, functional changes of *Kcnj13* occurred between *D. rerio* and *D. aesculapii*, possibly also between *D. rerio* and *D. tinwini* and *D. choprae*. However, we never observed pure *D. rerio Kcnj13* mutant patterns in

hemizygous hybrids, similar to mutant hybrids (Fig. 4l), indicating that the orthologs provide essential functions for patterning across all species tested and that a patterning function of *Kcnj13* might have predated the origin of the *Danio* genus. The separated positions of the three species with putative functional changes of *Kcnj13* in the phylogenetic tree (graph on the left of Fig. 6)[52] suggest a repeated and independent evolution of an ancestral gene function.

**The potassium channel gene _Kcnj13_.** Potassium channels have important roles in tissue patterning[59], notably in the regulation of allometric growth of fins in *D. rerio*[57,60–62]. *Kcnj13* encodes an inwardly rectifying potassium channel (Kir7.1) conserved in vertebrates (Supplementary Fig. 4). Mutations are known to cause defects in tracheal development in mice[63] and two rare diseases in humans leading to visual impairment[64–70]. During colour pattern formation in *D. rerio*, *Kcnj13* function is autonomously required in melanophores[35], and in ex vivo studies it was shown that the channel is involved in the contact-dependent depolarisation of melanophores upon interactions with xanthophores leading to a repulsion between these cells[39]. Evolution in *Kcnj13* in *D. aesculapii*, *D. tinwini* and *D. choprae* might therefore cause differences in heterotypic chromatophore interactions between species. The Kcnj13 protein functions as a tetramer, where each subunit contributes two transmembrane helices (M1 and M2, Supplementary Figs. 4 and 5) to the formation of the channel pore, as well as a short extracellular loop that folds back to form the pore lining ion selectivity filter (P-loop or H5, Fig. 4). The N- and C-termini of all four subunits reside in the cytoplasm, where they also contribute to the ion pore, but are mainly involved in gating of the channel (reviewed in Hibino et al.[71]). In *D. rerio* dominant mutant alleles of *Kcnj13* show broad stripes with irregular interruptions when heterozygous (Fig. 4i) and stronger pattern aberrations with fewer, wider and interrupted dark stripes and some mixing of melanophores and xanthophores when homozygous or trans-heterozygous. Three of them carry point mutations affecting H5 or M2, one is the result of a C-terminal truncation (Supplementary Fig. 5). The point mutations lead to proteins that do not produce functional channels and it has been suggested that the dominant phenotype is caused by a dosage-dependent effect, i.e., haploinsufficiency[35]. As the presumptive null allele we generated is recessive and shows a homozygous phenotype that is indistinguishable from the phenotype of the dominant alleles, these must in fact be dominant-negatives, where

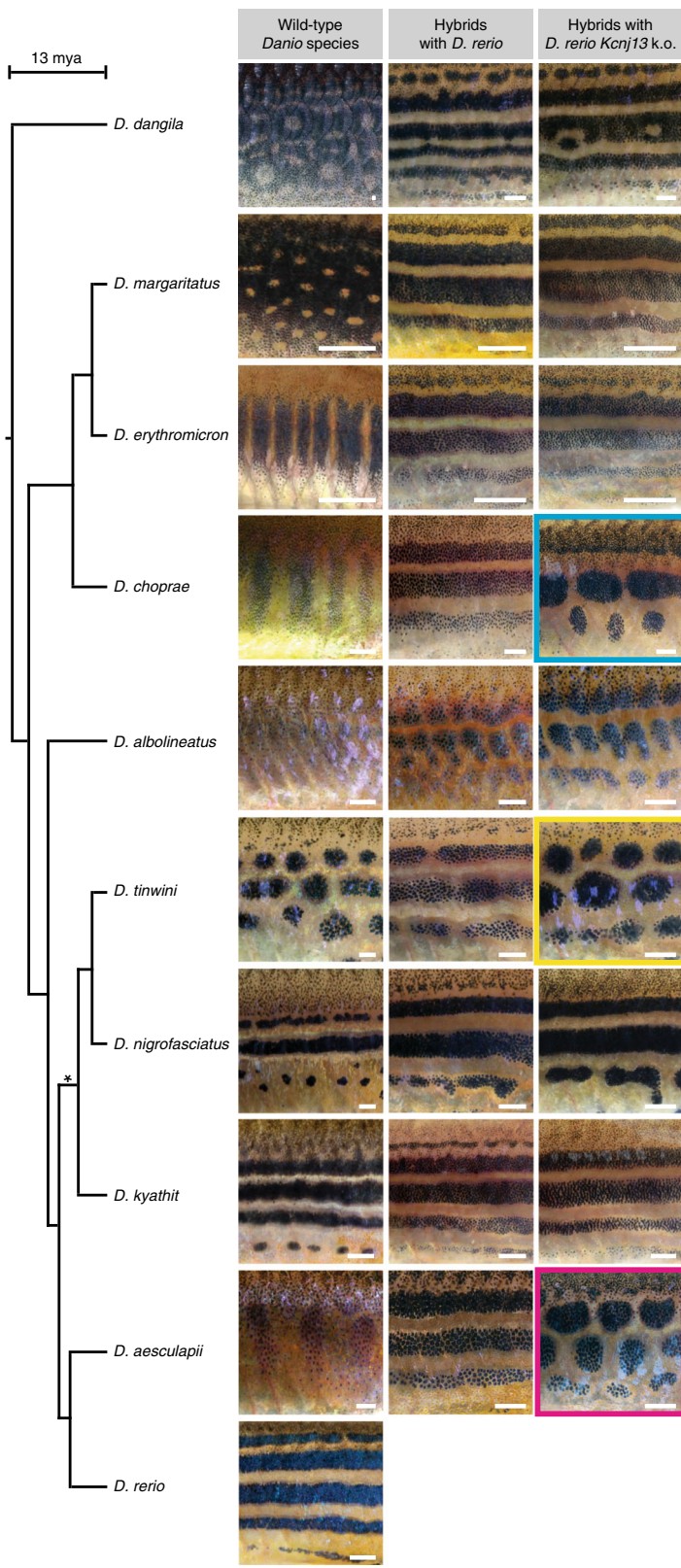

the mutant proteins inhibit the function of the wild-type protein still present in heterozygotes.

Using transcriptome data from species across the genus in combination with published data from *D. rerio*[72] we reconstructed the protein coding sequences of *Kcnj13* orthologs, which are highly conserved with only very few diverged sites in the cytoplasmic N- and C-terminal parts of the protein

(Supplementary Fig. 5). Whether these amino acid changes are the basis for the potentially repeated and independent evolution of *Kcnj13*, and if or how they might affect the function of the channel will require further experiments. The alleles in the three other species cannot simply be loss-of-function alleles, because the hybrid phenotypes differ from the homozygous mutant hybrids and also from *D. rerio Kcnj13* mutants. It is also possible

**Fig. 6 One-way complementation tests suggest repeated *Kcnj13* evolution.** On the left the phylogenetic tree depicts the relationship between the *Danio* species; the asterisk denotes a node with lower bootstrap support. The left column shows the patterns of the different species. In the middle column patterns of wild-type hybrids with *D. rerio* are shown (see also Supplementary Fig. 1). In the right column patterns of hybrids that carry a mutant *Kcnj13* allele from *D. rerio* are shown. Pattern defects are obvious in three cases: hybrids with *D. aesculapii* ($n > 60$, magenta), with *D. tinwini* ($n = 12$, yellow) and *D. choprae* ($n = 40$, cyan). In the other six cases the patterns in hemizygous hybrids do not differ from the striped patterns of wild-type hybrids (*D. rerio Kcnj13* k.o./*D. kyathit*, $n = 32$; *D. rerio Kcnj13* k.o./*D. nigrofasciatus*, $n = 16$, *D. rerio Kcnj13* k.o./*D. albolineatus*, $n = 4$; *D. rerio Kcnj13* k.o./*D. erythromicron*, $n = 38$; *D. rerio Kcnj13* k.o./*D. margaritatus*, $n = 12$; and *D. rerio Kcnj13* k.o./*D. dangila*, $n = 16$). All pictures show representative examples of the corresponding species/ hybrids/genotypes; for variability of the hybrid patterns see also Supplementary Fig. 3. Scale bars correspond to 1 mm.

that qualitative or quantitative changes in gene expression might be the basis for the observed divergence in gene function.

## Discussion

In contrast to mammals and birds, basal vertebrates retained several chromatophore types providing a substrate for the development of elaborate colour patterns. In zebrafish a relatively large number of genes regulating the formation of pigment patterns have already been identified by mutant screens. Due to the teleost-specific whole genome duplication and the following sub-functionalisation and retention of paralogs, many of these genes are specifically involved in adult colour patterning[14,15] and mutations in them show few, if any, pleiotropic effects[43,73]. Therefore, these genes are candidates for mediating pattern evolution[5,9].

In the *Danio* genus the pattern of *D. nigrofasciatus*, with fewer and interrupted dark stripes (Fig. 6), resembles the mutant phenotype of weak alleles involved in endothelin signalling in *D. rerio*. In zebrafish, endothelin signalling is directly required in iridophores for their development and proliferation; iridophores indirectly promote and sustain melanophore development[32]. Several paralogs exist for endothelin receptors and ligands[74], only one of each is specifically involved in patterning[7,29,73,75]. Indeed, it has been shown that in *D. nigrofasciatus*, due to cis-regulatory changes, the expression of the secreted ligand Endothelin 3b (Edn3b) is lower than in *D. rerio*[73]. Interspecific hybrids between the two species show lower expression of *Edn3b* from the *D. nigrofasciatus* allele compared to the *D. rerio* allele, confirming cis-regulatory changes in this gene between the two species.

We compared the development of the pattern in *D. rerio* with its closest sibling species, *D. aesculapii*, which has a completely different pattern of vertical bars. Whereas the orientation of the stripes in *D. rerio* depends on the presence of the horizontal myoseptum[32], as a structure through which the first iridophores reach the skin during metamorphosis, this is not the case in *D. aesculapii*. Here iridophores appear more scattered and only during later developmental stages (Fig. 2). This result might mean that in zebrafish the iridophores follow an attractive signal that lines the horizontal myoseptum; this signal, or the ability to respond to it, could be lost in *D. aesculapii*. The signal would be present in all the interspecific hybrids that have *D. rerio* as one parent, explaining the dominance of the horizontal stripes.

However, additional differences must also exist, because the pattern in *D. aesculapii* is very dissimilar to *D. rerio* mutants that lack the horizontal myoseptum (Fig. 1b)[32]. To address this question, we generated mutations in *D. aesculapii* that lead to the absence of one class of pigment cells (Fig. 3). The phenotypic analysis of these mutants showed that, if melanophores or xanthophores are missing, the remaining two cell types completely intermingle. This indicates that the cellular interactions are less complex in *D. aesculapii*. In contrast, we find that in the absence of iridophores a residual pattern is formed, which shows that iridophores, which play a leading role in patterning in *D. rerio*[20,32,33], only have a minor influence on the pattern in *D. aesculapii*.

To start revealing the genetic basis for the evolution of colour patterns in *Danio* fish we focused on a group of genes regulating heterotypic interactions among chromatophores[35–41]. These generally have strong recessive phenotypes and appear to have no obvious effects on other vital functions. Using reciprocal hemizygosity tests we identified the potassium channel gene *Kcnj13* as contributing to the patterning divergence between *D. rerio* and *D. aesculapii* (Fig. 5); one-way complementation tests suggest a broader role for *Kcnj13* in pattern diversification in the genus *Danio* including two more species, *D. tinwini* and *D. choprae* (Fig. 6). In *D. rerio* over 100 genes code for potassium channels of several different families, calcium-activated ($K_{Ca}$), two-pore ($K_{2P}$), voltage-gated ($K_V$) and inwardly rectifying ($K_{IR}$) channels. Potassium channels are expressed in many tissues and have diverse physiological roles, e.g., in the heart, kidney or nervous system. During development and regeneration potassium channels are involved in bioelectric signalling regulating allometric fin growth in *D. rerio*. Overgrowth of fins is caused by gain-of-function mutations in *Kcnh2a* (*longfin*)[62] and *Kcnk5b* (*another longfin*)[60]. In *schleier* mutants overgrowth is caused by a loss-of-function of the $K^+$-$Cl^-$-cotransporter Slc12a7a/Kcc4a[61]. It has also been shown that ectopic expression in the myotome of *Kcnj13* leads to overgrowing fins, arguing in favour of the predicted general role of this class of channels in setting the resting membrane potential of cells[57]. Zebrafish mutant for *Kcnj13*, including the newly generated null allele, are viable and show a phenotype specifically in pigment patterning; this might favour *Kcnj13* as a target for evolutionary change. The gene is expressed in other tissues besides chromatophores[76–78] and the apparent lack of pleiotropy could be due to redundancies with other potassium channels. *Kcnj13* is cell-autonomously required in melanophores, which appear ectopically in light stripes, and form irregular enlarged dark stripes sometimes intermixed with ectopic xanthophores in the mutants[35,37–39]. In *D. aesculapii*, mutations in *Kcnj13* cause a uniform distribution of melanophores, and no repulsive interactions with xanthophores are observed. The phenotype of hybrids with *D. aesculapii*, in which only the *D. aesculapii* allele is functional, is qualitatively different from the null allele of either species, and also from the dominant hypomorphic phenotype in *D. rerio*. This suggests that the change in *D. aesculapii* cannot simply be due to reduced expression levels, however spatial or temporal quantitative changes of gene expression might affect the function of the gene. Whether changes in the coding sequence are involved remains an open question. We do know, however, that *Kcnj13* from all tested species still has at least some residual function in patterning in the hybrids; none of them showed a complete mutant phenotype when only the *D. rerio* allele was non-functional. Therefore, we conclude that *Kcnj13* is active in colour pattern formation in all *Danio* species. Whereas the other patterning genes that we tested in *D. rerio*/*D. aesculapii* hybrids, *Cx39.4*, *Cx41.8* and *Igsf11*, show no divergence in function between these two species, it is likely that they are involved in pattern evolution in other species. The results of our study show that the genus *Danio* offers the opportunity to identify evolved genes and to reconstruct evolutionary history of biodiversity.

## Methods

No statistical methods were used to predetermine sample size. The experiments were not randomised. The investigators were not blinded to allocation during experiments and outcome assessment.

**Fish husbandry.** Zebrafish, *D. rerio*, were maintained as described earlier[79]. If not newly generated (Table 4 and Supplementary Information), the following genotypes were used: wild-type Tuebingen/TU, *nacre*^w2^/*Mitfa*[80], *pfeffer*^tm236^/*Csf1ra*[81], *transparent*^b6^/*Mpv17*[82], *leopard*^t1^/*Cx41.8*[37,41], *luchs* ^t37ui^/*Cx39.4*[40] and *obelix*^tXG6^/*Kcnj13*[40].

*D. aesculapii* and *D. albolineatus* were maintained identical to *D. rerio*. For the other *Danio* species, *D. kyathit*, *D. tinwini*, *D. nigrofasciatus*, *D. choprae*, *D. margaritatus*, *D. erythromicron* and *D. dangila* individual pair matings were not successful. Therefore, the fish were kept in groups in tanks containing boxes lightly covered with Java moss (*Taxiphyllum barbieri*), which resulted in sporadic matings and allowed us to collect fertilised eggs.

Interspecific hybrids were either obtained by natural matings or by in vitro fertilisations[54]. Hemizygous or homozygous mutant hybrids were identified by PCR and sequence analysis using specific primer pairs (Tables 1 and 3 and Supplementary Information).

All species were staged according to the normal table of *D. rerio* development[83]. All animal experiments were performed in accordance with the rules of the State of Baden-Württemberg, Germany, and approved by the Regierungspräsidium Tübingen.

**CRISPR/Cas9 gene editing.** The CRISPR/Cas9 system was applied either as described in Irion et al.[84] or according to the guidelines for embryo microinjection of Integrated DNA Technologies (IDT). Briefly, oligonucleotides were cloned into pDR274 to generate the sgRNA vector (Supplementary Tables 1 and 2). sgRNAs were transcribed from the linearised vector using the MEGAscript T7 Transcription Kit (Invitrogen). Alternatively, target-specific crRNAs and universal tracrRNAs were purchased from IDT. sgRNAs or crRNA:tracrRNA duplexes were injected as ribonucleoprotein complexes with Cas9 proteins into one-cell stage embryos. The efficiency of indel generation was tested on eight larvae at 1 dpf by PCR using specific primer pairs and by sequence analysis as described previously (Supplementary Tables 1 and 3)[85]. The remaining larvae were raised to adulthood. Mature F0 fish carrying indels were outcrossed. Loss-of-function alleles in heterozygous F1 fish were selected to establish homozygous or trans-heterozygous mutant lines (Supplementary Table 4).

**Image acquisition.** Anaesthesia of adult fish was performed as described previously[85]. A Canon 5D Mk II camera was used to obtain images. Fish with different colour patterns vary considerably in contrast, thus requiring different settings for aperture and exposure time, which can result in slightly different colour representations in the pictures. Juvenile fish were either embedded in low melting point agarose or fixed in 4% formaldehyde/0.08% glutaraldehyde and then photographed under a Leica MZ1 stereomicroscope (Fig. 2). Images were processed Adobe Photoshop and Adobe Illustrator CS6.

**Transcriptomics and sequence analysis.** Adult fish (*n* = 5 each for *D. rerio* (TU), *D. aesculapii*, *D. kyathit*, *D. nigrofasciatus*, *D. tinwini*, *D. albolineatus*, *D. choprae*, *D. erythromicron* and *D. margaritatus*) were euthanized by exposure to buffered 0.5 g/L MS-222 (Tricaine). Skin tissues were dissected in ice-cold PBS and collected using TRIzol (Life Technologies). RNA integrity and quantity were assessed by Agilent 2100 Bioanalyzer. Library preparation (TruSeq stranded mRNA, Illumina; 200 ng per sample) and sequencing (NovaSeq 6000, 2 × 100 bp) were performed by CeGaT GmbH (Tübingen, Germany). RNA-Seq analysis was carried out using the *Danio rerio* GRCz11 genome build for all *Danio* species and STAR aligner with default settings[86]. We found SNPs in the coding region of *Kcnj13* and considered other resources[87], including the latest zebrafish reference genome assembly (GRCz11), the ENA deposition Zebrafish Genome Diversity (PRJEB20043, Wellcome Trust Sanger) and the Zebrafish Mutation Project[72]. The variant calling pipeline for all *Danio* species consisted of GATK 3.8 and 4 and picard[88] from STAR-aligned bam files based on GATK Best-Practices pipeline. The full commands used can be found here: https://github.com/najasplus/STAR-deseq2 and https://github.com/najasplus/rnaseq_variant_calling. Variants were also called and checked using SAMtools, mpileup and bcftools[89]. The protein sequence alignment was produced using T-coffee[90].

**Reporting summary.** Further information on research design is available in the Nature Research Reporting Summary linked to this article.

## Data availability

The authors declare that all data supporting the findings of this study are available within the article and its supplementary information files or from the corresponding author upon reasonable request. The dataset generated during this study is available at The European Nucleotide Archive (ENA) accession number: PRJEB36360.

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

## Acknowledgements

We thank Silke Geiger-Rudolph, Horst Geiger and Roberta Occhinegro for excellent technical assistance and Patrick Müller for discussion. This work was supported by an ERC Advanced Grant "DanioPattern" (694289) and the Max Planck Society, Germany.

## Author contributions

All authors were involved in the design of the experiments. M.P., U.I. and H.G.F. performed the experiments. U.I., C.N.V., M.P., H.G.F. and C.M.D. analysed the data with support of A.E.; M.P. made the figures with contributions from U.I. and C.N.V.; U.I., C.N.V. and M.P. wrote the manuscript. C.N.V. and U.I. acquired funding.

## Funding

## Competing interests

The authors declare no competing interests.
