## [Peer Review File · Nature Communications]

Reviewers' Comments:

Reviewer #1:

Remarks to the Author:

This is an interesting study that has exploited the ability to generate CRISPR-induced mutations in many *Danio* fish species and inter-specific hybrids between these fish to identify genes that have contributed to natural variation in fish pigmentation patterns. In this paper, the authors (1) document diversity in pigmentation patterns among multiple *Danio* species and hybrids, (2) generate and perform phenotypic analysis of mutants for three genes in two species, (3) perform a reciprocal hemizyosity test for four genes between two of the species, and then (4) perform complementation tests between *D. rerio* mutants for *Kcnj13* and 8 other *Danio* species. Based on results of the reciprocal hemizyosity tests, the authors conclude that *Kcnj13* has evolved between *D. rerio* and *D. aesculapii* for pigmentation patterning and that three other genes have not. These are beautiful results that will be of considerable interest.

I have several comments that I hope will improve this manuscript and I use the author's helpful line numbering to organize my comments.

43 - The authors use the term complementation test to refer to the reciprocal hemizyosity test. I think it would be less confusing to use the commonly accepted term for these tests. The authors performed both reciprocal hemizyosity tests and one-way complementation tests. These two kinds of tests provide different kinds of information, which I will discuss below. Similarly, on line 85, the authors should distinguish between these two kinds of tests.

183-187. This paragraph is confusing. Perhaps should rewrite as "generate interspecific hybrids that carry loss-of-function" However, even with this grammatical change, the authors' conclusion is unclear to me. This is really a test of the remaining wild-type functions of genes in *rerio* versus hybrid. Indicates that genetic background behaves in similar way after loss of gene function. Not really sure what this experiment is telling us. Perhaps authors need to expand on this to clarify their meaning.

191-192 - revise - "loss-of-function allele from each parental species in an otherwise identical hybrid genetic background"

192-193 - Not strictly true that test was not performed previously in vertebrates. The first use was not as clean as in this paper. (Farahani P., Fidler J. S., Wong H., Diamant A. L., Yi N., Warden C. H., 2004 Reciprocal hemizyosity analysis of mouse hepatic lipase reveals influence on obesity. *Obes. Res.* 12: 292–305.) Perhaps best to just delete this sentence.

193-194 - revise - "We expect similar patterns in these hybrids if the gene function has not evolved between species." wild-type is not really a defined concept in these tests, since alleles from each species are presumably both wild-type, though they may have different functions.

196 - can also be gain of function.

Loss-of-function terminology is misleading in this evolutionary context.

Throughout, also important to clarify if these loss-of-function alleles are nulls. The reciprocal hemizyosity test is difficult to interpret if alleles are not nulls.

200-201 - rephrase - not a compensation issue. evolution of gene functions

202 - change to "hemizygous hybrids"

210 - would be helpful to be more specific: "...*Danio* genus, crossed the *D. rerio* *Kcnj13* null allele to seven other *Danio* species."

216 - This conclusion drawn from one-way hemizygous crosses is stronger than the results warrant. The observed phenotypes could, in principle, result from hemizyosity of *Kcnj13* in a novel hybrid background. This is why need the reciprocal hemizygotes. I would substantially soften these conclusions, "may" have evolved, but cannot rule out effect of hybrid background.

231 - "indicate" is too strong, given caveats mentioned above

Kcnj13 paragraph. The authors looked for potential coding changes and suggested that further work is required to follow up on possible effects of these coding differences. I don't know if *Kcnj13* is expressed specifically in chromatophores. But, if not expressed much in other tissues, then allele-specific qPCR in hybrids might be straightforward and would provide positive evidence that evolutionary change has resulted, at least in part, from changes of gene expression.

253-255 - This is a misinterpretation of evolutionary theory. In fact, both theory and empirical studies support the view that rapid evolution simply requires available genetic variability for the trait. This could be due to few or many loci. Some would argue that variation at multiple loci would allow for more rapid evolution than variation at few loci. In any case, I don't think this sentence is clarifying and recommend cutting.

255-256 - This is the more interesting evolutionary result. Why do the same genes evolve repeatedly? Maybe the authors can speculate about why *Kcnj13* might be a favored node of evolutionary change for this kind of phenotypic evolution. Stern & Orgogozo 2009 (Is genetic evolution predictable? *Science* 323: 746–751.) discuss how the structure of developmental networks can bias genetic evolution. Can the authors provide any insight into how the structure of the pigmentation patterning network might influence genetic evolution?

478 - Would be nice to show data confirming loss of function alleles. Are these nulls? Really want nulls for RHT.
RT-PCR, Northernblots? Westerns?

500 - This differential expression analysis is not mentioned in text.

514 - This sequence analysis is not discussed.

620 - Perhaps enough room in each panel to spell out full species names

648 - Would be nice to see examples of more individuals from RHT, perhaps in supp material? Evolutionary geneticists will not be convinced by images of single fish.

David L. Stern (Feel free to contact me directly if you want to discuss interpretation of RHT and one-way complementation tests.)

Reviewer #2:

Remarks to the Author:

The authors investigate the genetic basis for skin pattern variation across species in the *Danio* genus. Several results are outlined in the paper. First, some of the differences between *D. rerio* and *D. aesculapii* patterning are attributed to different cellular interactions between (1) melanophores and xanthophores, and (2) iridophores and other chromatophores; the authors also provide further evidence that the mechanisms behind stripe formation in *D. rerio* are partially redundant (so double mutants are needed to change patterns), while single mutants in *D. aesculapii* already exhibit changed patterns. The main result is about *Kcnj13*: using the

CRISPR/Cas9 system and complementation tests, the authors demonstrate that Kcnj13 evolved across several species.

These results are novel and interesting, and I found the arguments and methodologies convincing. I believe that this manuscript will be of significant interest to the zebrafish community and inspire future work. I have a few suggestions for the authors:

1. It would help if the authors could provide a short summary of their main findings that is separate from the experimental details. Most paragraphs contain some of these conclusions, and it would help if readers could find them in a central place in the manuscript.

2. How many fish were used in the different experiments that the authors conducted? How variable were the patterns across these fish? I believe it would be helpful to get a better sense of variability of patterns in these studies.

3. Minor comments:

Line 479: Reference is missing in "(Ref)"

Line 529: accession number: replace XXX by pending?

Line 673: box for *D. rerio* Kcnj13 k.o./*D. aesculapii* is purple, not green

Point-by-point response to REVIEWERS' COMMENTS

Reviewer #1:

43 - The authors use the term complementation test to refer to the reciprocal hemizyosity test. I think it would be less confusing to use the commonly accepted term for these tests. The authors performed both reciprocal hemizyosity tests and one-way complementation tests. These two kinds of tests provide different kinds of information, which I will discuss below. Similarly, on line 85, the authors should distinguish between these two kinds of tests.

We have revised the manuscript according to this comment to distinguish more clearly between reciprocal hemizyosity tests and complementation tests; also, with regards to the kind of information each of these tests can provide.

183-187. This paragraph is confusing. Perhaps should rewrite as "generate interspecific hybrids that carry loss-of-function" However, even with this grammatical change, the authors' conclusion is unclear to me. This is really a test of the remaining wild-type functions of genes in *rerio* versus hybrid. Indicates that genetic background behaves in similar way after loss of gene function. Not really sure what this experiment is telling us. Perhaps authors need to expand on this to clarify their meaning.

We have re-written this paragraph in order to make this point clearer: This experiment is telling us that mutations in the same genes affect pattern formation in the hybrids in a way that is very similar to stripe formation in *D. rerio*, thus, the mutant hybrids shows a (predicted) phenotype similar to homozygous mutants in *D. rerio*. Therefore, we conclude that the mechanisms are comparable and the novel phenotype we find in the hemizygous hybrids (*Kcnj13* loss from *D. rerio*) is indeed novel, and it indicates that the allele from *D. aesculapii* functions in patterning.

191-192 - revise - "loss-of-function allele from each parental species in an otherwise identical hybrid genetic background"

We have revised this paragraph to better convey the rationale for our experiments and the conclusions we draw from them.

192-193 - Not strictly true that test was not performed previously in vertebrates. The first use was not as clean as in this paper. (Farahani P., Fisler J. S., Wong H., Diament A. L., Yi N., Warden C. H., 2004 Reciprocal hemizyosity analysis of mouse hepatic lipase reveals influence on obesity. *Obes. Res.* 12: 292–305.) Perhaps best to just delete this sentence.

Thanks for pointing this out; we had missed this paper. We deleted the sentence.

193-194 - revise - "We expect similar patterns in these hybrids if the gene function has not evolved between species." wild-type is not really a defined concept in these

tests, since alleles from each species are presumably both wild-type, though they may have different functions.

We agree that wild-type is not a good concept to describe differences in gene function between species; we have revised the sentence to avoid these difficulties.

196 - can also be gain of function.

Loss-of-function terminology is misleading in this evolutionary context.

Again, we agree, loss- or gain-of-function terminology can be misleading in an evolutionary context. Therefore, we have changed this in the manuscript, and made additional changes to emphasize that all non-mutant alleles from any species are wild-type (see previous point).

Throughout, also important to clarify if these loss-of-function alleles are nulls. The reciprocal hemizyosity test is difficult to interpret if alleles are not nulls.

This is indeed a very important point. Although we have no absolute proof that the loss-of-functions alleles we generated with the CRISPR/Cas system are nulls, all the evidence we have points towards that. In all cases we selected for frame-shift mutations occurring early in the coding sequence, which are predicted to lead to truncated proteins that are non-functional. In the case of *Igsf11* in *D. rerio* this produced a stronger phenotype than the previously known mis-sense alleles, indicating that we produced indeed a stronger allele. In the other cases from *D. rerio* we never observed any phenotype that might lead to the conclusion the alleles might not be nulls. Both connexins produce the expected spotted phenotypes when mutated, for *Kcnj13* we find that the frame-shift leads to a recessive allele, which, when homozygous, produces a phenotype indistinguishable from the homozygous or trans-heterozygous mutant dominant alleles. In the other species, *D. aesculapii*, again, we don't have absolute proof that the mutations lead to null-alleles, but we don't find any inconsistencies in the phenotypes looking at different alleles from the same gene. And we find that several of the knock-outs lead to more severe phenotypes, i.e. complete loss of patterning, than might have been expected from solely looking at the corresponding mutants in *D. rerio*. Taken together, we think that all the evidence points towards the fact that the alleles we're working with are functional null-alleles.

200-201 - rephrase - not a compensation issue. evolution of gene functions

Thanks for pointing this out; we changed the sentence.

202 - change to "hemizygous hybrids"

done

210 - would be helpful to be more specific: "...Danio genus, crossed the *D. rerio* *Kcnj13* null allele to seven other Danio species."

We have changed this sentence accordingly.

216 - This conclusion drawn from one-way hemizygous crosses is stronger than the results warrant. The observed phenotypes could, in principle, result from

hemizygosity of *Kcnj13* in a novel hybrid background. This is why need the reciprocal hemizygotes. I would substantially soften these conclusions, "may" have evolved, but cannot rule out effect of hybrid background.

We agree that conclusions from one-way complementation crosses need to be drawn carefully. Therefore, we revised the manuscript to soften these.

231 - "indicate" is too strong, given caveats mentioned above

same as above

Kcnj13 paragraph. The authors looked for potential coding changes and suggested that further work is required to follow up on possible effects of these coding differences. I don't know if *Kcnj13* is expressed specifically in chromatophores. But, if not expressed much in other tissues, then allele-specific qPCR in hybrids might be straightforward and would provide positive evidence that evolutionary change has resulted, at least in part, from changes of gene expression.

We know that *Kcnj13* is not only expressed in chromatophores but also in other cells. It was recently published that during early larval stages it is also expressed in the pronephric duct (Silic et al., (2020) *Genetics* **215** (4), 1067-1084). For adult fish we have evidence from RNAseq experiments that expression levels in the skin are not dramatically altered even if all chromatophores are absent. We have performed more RNAseq experiments to test for allele-specific expression in the hybrids; preliminary results indicate that expression levels from the *D. aesculapii* allele are reduced in the hybrids. However, a more thorough analysis of the results from these experiments is still ongoing and will take too long to be included in this publication. In addition, we also have preliminary evidence from transgenic experiments indicating that the coding sequences from *D. rerio* and from *D. aesculapii* are not equally good in rescuing the mutant phenotype in *D. rerio*. Therefore, a combination of coding sequence changes and regulatory changes might underlie the divergence of the genes between *D. rerio* and *D. aesculapii*; something we plan to investigate in the future in more detail.

253-255 - This is a misinterpretation of evolutionary theory. In fact, both theory and empirical studies support the view that rapid evolution simply requires available genetic variability for the trait. This could be due to few or many loci. Some would argue that variation at multiple loci would allow for more rapid evolution than variation at few loci. In any case, I don't think this sentence is clarifying and recommend cutting.

We cut this sentence from the discussion.

255-256 - This is the more interesting evolutionary result. Why do the same genes evolve repeatedly? Maybe the authors can speculate about why *Kcnj13* might be a favored node of evolutionary change for this kind of phenotypic evolution. Stern & Orgogozo 2009 (Is genetic evolution predictable? *Science* 323: 746–751.) discuss how the structure of developmental networks can bias genetic evolution. Can the authors provide any insight into how the structure of the pigmentation patterning network might influence genetic evolution?

We are seemingly just at the beginning of unraveling the structure of a potential pigmentation patterning network, still with many unknowns. Therefore, the structure of the network isn't quite clear, yet. What we can say is, that despite the wider expression of *Kcnj13*, mutations do not have pleiotropic effects, which might be due to redundancies in other tissues that express other potassium channels. This maybe one reason for repeated evolution. We have added these points to the discussion.

478 - Would be nice to show data confirming loss of function alleles. Are these nulls? Really want nulls for RHT.
RT-PCR, Northern? Westerns?

Unfortunately, we don't have antibodies for Western Blots to demonstrate the absence of the proteins. We tried RT-PCRs, but the results are inconclusive, so far. However, we don't know if mRNA stability is affected in the mutants. For the connexins, which contain the entire coding sequence in one exon, we do not expect nonsense-mediated mRNA decay; for the other genes we don't know to what extent it might occur. For this reason, we did not perform Northern Blots.

However, as described above, we selected for frame-shift mutations that are predicted to lead to a complete loss of the gene function; and, we don't have any experimental evidence that this might not be the case.

500 - This differential expression analysis is not mentioned in text.

We have removed this from the methods part, because it is indeed not discussed in the manuscript, and the sequences we got from this RNAseq experiment are still valid without the differential expression analysis (see also point below).

514 - This sequence analysis is not discussed.

We performed several RNAseq experiments; the data of most are still being analyzed. However, those experiments are the sources for the *Kcnj13* sequences from the different *Danio* species we present in this manuscript. Therefore, we feel it is necessary to mention these experiments in the Methods part.

620 - Perhaps enough room in each panel to spell out full species names

We have changed the figures accordingly.

648 - Would be nice to see examples of more individuals from RHT, perhaps in supp material? Evolutionary geneticists will not be convinced by images of single fish.

We have added more examples of the hybrid fish in Figures S1 – S3 to show the variability (or lack thereof) of the pigmentation phenotypes.

Reviewer #2:

1. It would help if the authors could provide a short summary of their main findings that is separate from the experimental details. Most paragraphs contain some these conclusions, and it would help if readers could find them in a central place in the manuscript.

We have revised the manuscript and added a short summary paragraph at the end of the introduction.

2. How many fish were used in the different experiments that the authors conducted? How variable were the patterns across these fish? I believe it would be helpful to get a better sense of variability of patterns in these studies.

We have added the number of fish that were analyzed in the different experiments to the appropriate figure legends. We have also added several examples of the hybrid fish in three supplementary figures to show the variability of the pigmentation patterns in the hybrids.

3. Minor comments:

Line 479: Reference is missing in "(Ref)"

Thanks, we have changed that.

Line 529: accession number: replace XXX by pending?

We have an accession number and it is now mentioned in the Methods part.

Line 673: box for *D. rerio* Kcnj13 k.o./*D. aesculapii* is purple, not green

We apologize for this mistake; it has been changed.

Reviewers' Comments:

Reviewer #1:

Remarks to the Author:

The authors have done a nice job addressing my initial concerns. I noticed several small things they may want to modify or delete.

line 62 - ddp? I think they mean dpp.

line 138 - not quite sure what "over" means in this phrase
"over vertical bars in *D. aesculapii*"

line 215-218 - "These hybrid patterns are more variable than the species patterns (Fig. S1) suggesting that the ancestral patterns did not function as recognition signals but rather provided camouflage." - I do not understand the logic of this sentence. It is very common to observe that phenotypes are more variable in hybrids than pure species. Sturtevant recognized this in *Drosophila mel* X *sim* hybrids 100 years ago and posited, correctly I think, that this reflects incompatibilities between the species.

Reviewer #2:

Remarks to the Author:

The authors addressed my questions and suggestions satisfactorily. As I wrote in my first report, the results obtained by the authors in this manuscript are novel and interesting, and I found the arguments and methodologies convincing. I believe that this manuscript will be of significant interest to the zebrafish community and inspire future work.